# Towards Building a Trustworthy Deep Learning Framework for Medical Image Analysis

**DOI:** 10.3390/s23198122

**Published:** 2023-09-27

**Authors:** Kai Ma, Siyuan He, Grant Sinha, Ashkan Ebadi, Adrian Florea, Stéphane Tremblay, Alexander Wong, Pengcheng Xi

**Affiliations:** 1Faculty of Engineering, University of Waterloo, Waterloo, ON N2L 3G1, Canada; sy4he@uwaterloo.ca (S.H.); ashkan.ebadi@nrc-cnrc.gc.ca (A.E.); a28wong@uwaterloo.ca (A.W.); 2Digital Technologies Research Centre, National Research Council of Canada, Ottawa, ON K1A 0R6, Canada; stephane.tremblay@nrc-cnrc.gc.ca; 3Faculty of Mathematics, University of Waterloo, Waterloo, ON N2L 3G1, Canada; gsinha@uwaterloo.ca; 4Department of Emergency Medicine, McGill University, Montreal, QC H4A 3J1, Canada; adrian.florea@mail.mcgill.ca

**Keywords:** computer-aided diagnosis, COVID-19, feature learning, self-supervised learning, contrastive learning, AUC, trustworthiness

## Abstract

Computer vision and deep learning have the potential to improve medical artificial intelligence (AI) by assisting in diagnosis, prediction, and prognosis. However, the application of deep learning to medical image analysis is challenging due to limited data availability and imbalanced data. While model performance is undoubtedly essential for medical image analysis, model trust is equally important. To address these challenges, we propose TRUDLMIA, a trustworthy deep learning framework for medical image analysis, which leverages image features learned through self-supervised learning and utilizes a novel surrogate loss function to build trustworthy models with optimal performance. The framework is validated on three benchmark data sets for detecting pneumonia, COVID-19, and melanoma, and the created models prove to be highly competitive, even outperforming those designed specifically for the tasks. Furthermore, we conduct ablation studies, cross-validation, and result visualization and demonstrate the contribution of proposed modules to both model performance (up to 21%) and model trust (up to 5%). We expect that the proposed framework will support researchers and clinicians in advancing the use of deep learning for dealing with public health crises, improving patient outcomes, increasing diagnostic accuracy, and enhancing the overall quality of healthcare delivery.

## 1. Introduction

Medical imaging plays a key role in dealing with health problems on a daily basis. To analyze large volumes of imaging data, computer-assisted diagnosis has shown great potential through computer vision and deep learning [1,2]. In particular, deep learning models can be trained to identify subtle changes in medical imaging data that indicate the onset of a disease, allowing for earlier intervention and better outcomes. In the case of a pandemic, medical imaging can be used to identify and track the spread of the disease, and deep learning can be used to identify patterns from medical imaging data and help predict the course of the disease. Nevertheless, three main issues have been identified in the effective use of deep learning for medical applications.

**Limited data.** Datasets for medical imaging, especially for novel diseases, are typically small compared to natural image datasets, making model training more difficult. As such, medical feature learning is commonly conducted through transfer learning [3], with large-scale supervised learning followed by downstream fine-tuning. This approach, however, is limited by the relevance of the large-scale data to the downstream task, as well as label quality. Consequently, self-supervised learning (SSL) has been proposed and has shown comparable performance to state-of-the-art supervised models [4,5]. Contrastive self-supervised learning, which aims to learn feature representations by comparing closely related data samples to each other, is especially relevant. In this method, extensive data augmentation is utilized to improve the efficiency of feature learning, thus requiring less data and labeling.

**Class imbalance.** Aside from small datasets, another issue for medical imaging is class imbalance, where there are significantly more negative (benign) data samples than positive (malignant) ones. Thus, models are heavily biased toward the majority negative class and exhibit poor predictive performance for the minority positive class, which if misdiagnosed, leads to serious consequences. This issue is especially serious for classifiers trained with traditional loss functions like stochastic gradient descent [6]; existing methods for dealing with class imbalance also have significant weaknesses [7,8,9] that make them unsuitable for medical use cases. A way to combat this during the model training process is to maximize the Area under the receiver operating characteristic (AUC) instead of minimizing cross-entropy (CE) loss [10]. This is suitable for imbalanced data, as maximizing AUC aims to rank the prediction score of positive samples higher than negative ones. However, AUC maximization is more sensitive to model changes, making training less practical [10].

**Low trustworthiness.** In the context of medical artificial intelligence (AI), the trustworthiness of predictions is important to both patients and clinicians. A prevalent problem is that deep neural networks optimized with the standard CE loss function tend to be overly cautious for the minority class, while being overconfident for the majority class [6]. This problem is especially hard to deal with, as model trust quantification is a relatively new and under-developed area. To deal with limitations in the existing literature on model trustworthiness, a concept of “question–answer” trust has been introduced [11], where the trustworthiness of a model is determined by its behavior when answering questions, such that undeserved confidence is penalized while well-placed confidence is rewarded. Through this method, a simple scalar “trust score” is introduced for quantitative assessment of model trustworthiness.

To address the aforementioned issues, we propose **TRUDLMIA**, a trustworthy deep learning framework for medical image analysis. Both supervised and self-supervised learning are combined for effective medical image feature learning and a novel surrogate AUC margin loss function is adopted to build high-performing and high-trust models. Weaknesses of each module are supplemented by other modules to reduce limitations such as high computational resource requirements for self-supervised learning and training-time instability with AUC margin loss. Moreover, the framework adopts a model-agnostic modular design for generalization capabilities.

In summary, our contributions and findings are three-fold:We propose a general deep learning framework for medical image analysis which can be used to build high-performing and high-trust models;We demonstrate that models built using the proposed framework outperform existing ones on benchmark tasks. While the main focus of this work is COVID-19 diagnosis, we verify the generalization ability of our framework by achieving excellent results on pneumonia and melanoma classification tasks, outperforming models specifically designed for those tasks;We study the effect of AUC maximization with margin loss on model trustworthiness, with the goal of addressing under/over-confidence caused by class imbalance.

## 2. Relevant Work

In computer vision, self-supervised learning (SSL) has gained increasing popularity for learning visual representations. Overall, the SSL approaches can be categorized into generative or discriminative. Momentum Contrast (MoCo) [5] and SimCLR [4], two main-stream discriminative approaches, learn feature encoders by comparing data samples to each other. MoCo approaches this task through a mechanism analogous to dictionary look-up. Through a contrastive loss function, a visual representation encoder is trained by matching encoded queries to a dictionary of encoded keys. SimCLR, however, focuses on the use of data augmentations on the same image. During its training phase, SimCLR utilizes a contrastive loss function to enhance the representational similarity between different augmentations of the same image (positive pairs), while concurrently amplifying the dissimilarity between the representations of distinct images (negative pairs).

Contrastive learning methods have been shown to be effective in medical contexts. In [12], the MoCo model is pre-trained on a chest X-ray data set to produce the MoCo-CXR model. Subsequent end-to-end fine-tuning experiments show that models initialized with MoCo-CXR outperformed their counterparts, especially on limited training data. By experimenting on unseen datasets, it also shows that MoCo-CXR pre-training has good transferability across chest X-ray datasets and tasks.

Self-supervised learning with MoCo has also been used for tackling pandemics. In [13], the authors pre-train the MoCo model on chest X-rays to learn more general image representations to use for prognosis tasks, including adverse event deterioration and increased oxygen requirements. This method proves effective in that existing solutions leverage supervised pre-training on non-COVID images, an approach limited by the differences between the pre-training data and the target COVID-19 patient data. It thus achieves comparable prediction accuracy to that of experienced radiologists analyzing the same information.

SimCLR has also been previously applied in medical contexts. In [14], the authors study the effectiveness of self-supervised contrastive learning as a pre-training strategy within the domain of medical image classification. They propose Multi-instance Contrastive Learning (MICLe), a novel approach that generalizes contrastive learning to leverage special characteristics of medical image analysis. They observe that the SSL pre-trained on ImageNet, followed by additional pre-training on unlabeled domain-specific medical images, improves the accuracy of medical image classifiers [14].

Addressing the issue of imbalanced data in machine learning is of significant importance to ensure unbiased and accurate predictions. Many traditional methods exist for the purpose of tackling this issue. One popular approach is oversampling, which involves artificially inflating the minority class by replicating instances or introducing synthetically generated examples, a technique demonstrated by methods such as the Synthetic Minority Over-sampling Technique (SMOTE) [15]. Conversely, undersampling reduces the size of the majority class to balance the data. Though effective, these resampling techniques might lead to overfitting (in the case of oversampling) or loss of potentially important information (in undersampling) [16]. Another approach is the use of cost-sensitive learning, which adjusts the cost function to make misclassification of the minority class more penalizing [8]; however, determining appropriate costs is difficult, and sub-optimal cost values may result in overfitting the minority class or ignoring the majority class [17]. Lastly, ensemble methods, such as Random Forests or boosting algorithms, have proven to be effective in dealing with imbalanced data due to their inherent ability to handle bias towards the majority class [7,18]. The main weakness of these ensemble methods lies in their requirement for multiple base learners, which can lead to high computational costs. This makes them slower to train and more resource-intensive than single models [9]. Moreover, the interpretability of models can be an important contributing factor to model trustworthiness, which is important for medical applications; thus, the uninterpretable black-box nature of decision trees [19] presents significant inherent limitations.

In medical image analysis, AUC has been widely utilized for performance evaluation. The recent proposal of deep AUC maximization [10], a novel surrogate loss used instead of standard CE loss, aims to directly optimize the AUC metric. This is especially beneficial for medical data where the positive class is a minority, as the AUC metric ranks the prediction score of positive samples higher than negative ones. However, directly maximizing AUC score is challenging. Thus, the authors propose a new margin-based min-max surrogate loss function for the AUC score, which proves to outperform CE loss by a large margin [10] and is robust to noisy data. The use of this surrogate loss function has led to large increases in network performance. Moreover, the use of AUC maximization with supervised models has achieved great success in winning the Stanford CheXpert competition [20].

Model trustworthiness quantification is a relatively new and under-developed area compared to other deep learning performance metrics, such as robustness [21], efficiency [22], or explainability [23]. Existing work on quantifying model trust typically focuses on evaluating trustworthiness for a prediction made on a single data sample, either through measuring agreement with a nearest-neighbor classifier [24] or estimations for model uncertainty [25]. Unfortunately, these approaches often suffer from weaknesses due to returning distributions over the space of possible predictions. Their trust quantification is often highly complex, hard to interpret, or limited to Bayesian neural networks [26,27]. To deal with these limitations, a concept of “question–answer” trust is introduced in [11], where the trustworthiness of a model is determined by its behavior when answering questions correctly or incorrectly—undeserved confidence is penalized while well-placed confidence is rewarded. Through this method, a simple scalar “trust score” is introduced to express question–answer to the practice: a higher trust score indicates a more trustworthy model.

Because of the proven advantage of SSL pre-training and AUC maximization, our proposed approach extends previous work [28] to combine it with self-supervised pre-training, AUC maximization, and easily interpretable model trust quantification. We aim to build a deep learning framework to maximize model performance and trustworthiness in the context of tackling public health crises.

## 3. Materials and Methods

### 3.1. Framework Introduction

The TRUDLMIA framework comprises three modules: (i) generic learning through large-scale supervised pre-training using natural images (ImageNet [29]), (ii) adapted learning through large-scale self-supervised learning using natural images or domain images without labels, and (iii) targeted learning through supervised fine-tuning on downstream tasks using a labeled dataset. The framework is shown in Figure 1, where the three modules are built upon each other to create trustworthy models with optimal performance, despite the fact that training data on target tasks are limited in quantity and imbalanced.

The design of the proposed framework is based on the following considerations. Conducting supervised pre-training (module (i)) first leads to more efficient SSL training as the latter can be very costly. For module (ii), we study two main-stream SSL approaches, namely SimCLR and MoCo, on their performance and trustworthiness as pre-training for our framework. In the comparison below and in ablation studies, we mainly use SimCLR, as we found that it performed better than MoCo for the given task (results shown in Section 4.3). We use AUC maximization with AUC min-max margin loss in module (iii) of the TRUDLMIA framework, which has several benefits over traditional cross-entropy (CE) loss. Most importantly, AUC maximization is better at handling imbalanced data [10], being more resistant to trust issues caused by CE loss (over-confidence in the majority class and over-cautiousness in the minority class [6]).

The TRUDLMIA framework adopts a plug-in architecture in the computation of image features. In our study, we adopt main-stream deep Convolutional Neural Network (CNN) models, i.e., ResNet [30] and DenseNet [31], which can be replaced with other network architectures. The two self-supervised learning (SSL) approaches compared in module (ii) are also replaceable. Likewise, the AUC maximization used in module (iii) can be replaced with alternative loss functions.

### 3.2. Data

Different datasets are used for pre-training and fine-tuning TRUDLMIA modules. In supervised pre-training (module i), the deep CNN models are pre-trained on the ImageNet [29] dataset. In self-supervised learning (module ii), the MoCo model is pre-trained on both ImageNet and MIMIC-CXR dataset [32], whereas the SimCLR model is pre-trained on the ImageNet dataset only.

For downstream tasks (module iii), the COVIDx dataset [33], a small dataset with a high class imbalance, is first used for fine-tuning the models end-to-end. The dataset training/testing split is shown in Table 1. In order to test the generalization capability of the TRUDLMIA framework, the RSNA Pneumonia Challenge dataset [34] and the SIIM-ISIC Melanoma Classification dataset [35] are used to validate our results. These datasets are also small with a high class imbalance (see Table 2 and Table 3).

### 3.3. Trust Score Computation

We compute trust scores for models based on the methods introduced in [11]. Given a question *x*, an answer *y* with respect to a model *M*, such that y=M(x), and *z* representing the correct answer to *x*, we use $R_{y=z|M}$ to denote the space of all questions where the answer *y* given by model *M* matches the correct answer *z*. Likewise, we use $R_{y\neq z|M}$ to denote the space of all questions where the answer *y* given by the model does not match the correct answer. We also define the confidence of *M* in an answer *y* to question *x* as C(y∣x). Thus, **question–answer trust** of an answer *y* given by model *M* of a question *x*, with knowledge of the correct answer *z*, is defined as
$$Q_{z}\left(x,y\right)=\left\{\begin{array}{cc} C{\left(y|x\right)}^{\alpha }, & \mathrm{if}\;x\in R_{y=z|M}\\ {\left(1-C\left(y|x\right)\right)}^{\beta }, & \mathrm{if}\;x\in R_{y\neq z|M}, \end{array}\right.$$
with α and β denoting reward and penalty relaxation coefficients.

To put the approach into practical usage with our models, we first calculate an optimal threshold value by maximizing the F1-score on the validation split. Data samples are passed through the model and outputs lower than the threshold are predicted to be negative, while outputs above the threshold are predicted to be positive. Model outputs are then normalized, such that negative predictions are scaled between 0 and 0.5 and positive predictions are scaled between 0.5 and 1. This allows us to express model confidence, which is then used to compute a **trust score** according to the question–answer method introduced above. We use α=1 and β=1, equally rewarding well-placed confidence and undeserved overconfidence. This is performed for all of the positive samples in the unseen test split. Finally, an overall positive class trust score for the model is determined by calculating the mean of the computed individual scores.

### 3.4. Model Selection

During fine-tuning, validation is conducted on a randomly sampled validation split (10% of the original training set). In order to construct the validation set, we adopt a balanced random sampling strategy. This approach guarantees a random selection of instances while maintaining an equal representation from each class. This procedure aligns the validation set’s distribution with that of the test set, thereby facilitating a more precise estimation of model performance during the validation phase.

For each training session, we save the following models: best validation accuracy, best validation AUC score, best validation F1 score, lowest validation loss, and last epoch. However, when comparing different model architectures, learning methods, or loss functions, we strictly use models saved under the “best validation F1 score” criteria for the sake of consistency. This criterion combines both precision and recall, which are important metrics commonly used in the medical context.

## 4. Results

### 4.1. Framework Validation

We implemented the proposed TRUDLMIA framework for validation on three distinct medical datasets: COVIDx 8B, RSNA Pneumonia Challenge, and SIIM-ISIC Melanoma Classification.

Our top-performing model on the COVIDx dataset, COVID-CXR-SSL, showcased exceptional performance and trust score (as demonstrated in Table 4). The model incorporates ResNet for the first module, SimCLR for the second, and subsequently fine-tunes the COVIDx dataset during the third module.To benchmark the performance of COVID-CXR-SSL, we selected two state-of-the-art models, COVID-Net CXR-2 [36] and COVID-Net CXR-3 [37], that have been specifically designed and optimized for the same dataset. The former utilizes machine-driven design to autonomously discover highly tailored macro/micro-architecture designs, while the latter employs a self-attention mechanism (MEDUSA). Both these models have set high benchmarks across various medical imaging tasks. Notably, COVID-CXR-SSL surpassed the performance of both COVID-Net CXR-2 and COVID-Net CXR-3 across all evaluation metrics on the COVIDx V8B dataset. Despite operating on a lower resolution of 224 × 224 compared to the 480 × 480 utilized by the COVID-Net CXR models, our model still demonstrated superior performance.

Similarly, we apply the TRUDLMIA framework to create and train a model on the RSNA Pneumonia Challenge dataset. This is compared against other notable models, achieving top performance in 2 categories (shown in Table 5). Please note that we chose different metrics used in this comparison, as we are matching the metrics used by other publications to produce a fair comparison. Our model, dubbed RSNA-Pneumonia-SSL, achieves superior performance to other known models on most metrics [37].

Finally, to further verify the generalization ability of the model, we expand outside of the chest radiograph domain and train a model on SIIM-ISIC Melanoma Classification dataset, using the TRUDLMIA framework. Once again we match the metric for comparison to correspond to the metric used in the challenge; the main metric used by the SIIM-ISIC Melanoma Detection Challenge is the AUC score. Our model, named TRUDLMIA-Melanoma, achieves top 1% performance. This is compared against other notable models, achieving competitive performance (shown in Table 6). As shown in the table, many of the other high-performing models use various methods and useful tricks to improve their final result; most notably, many of them use external data collected from other datasets, or heavily use data augmentations. Furthermore, many other submissions use model ensembles. This can be especially useful for improving performance as shown by the winning submission [41], which uses an 18-model ensemble and uses multiple backbones. On the other hand, TRUDLMIA achieves competitive performance with just a single model, no external data, and no data augmentations. We also outperform other single-model submissions [42] and AUC maximization-based submissions [10].

### 4.2. Ablation Study

We conduct ablation studies to investigate the contribution of different modules in the TRUDLMIA framework. We start with fine-tuning a pre-trained ResNet on the COVIDx dataset as a baseline (model “SL”). The pre-trained ResNet model is also used as a backbone architecture for training with the SimCLR architecture followed by fine-tuning it using the CE loss function (model “SL + SSL”). Furthermore, the SimCLR model is also fine-tuned using the AUC maximization loss function (model “SL + SSL + AUC”). Both the “SL + SSL” and “SL + SSL + AUC” models are fine-tuned for 200 epochs. Table 7 lists the performance metrics and trust scores computed on the models. We obtain an increase of about 6% on precision and sensitivity metrics and 4% on trust score from the adoption of SSL. AUC maximization further improves the performance while improving the trust score slightly.

To verify the ablation study, we repeat it on the RSNA Pneumonia Challenge dataset (results in Table 8). The introduction of the SSL and AUC modules improves all metrics, with the most notable ones the sensitivity of the positive class at around 21% and model trust at around 5%.

### 4.3. Selection of SSL Plug-In

Given the choice of different SSL approaches, we conduct a comparison with MoCo architecture in the TRUDLMIA framework. After fine-tuning for 100 epochs, we select the best models for evaluation and provide their performance metrics in Table 9. The SimCLR based model outperforms the MoCo-based one across all metrics. Our results also indicate that pre-training with natural images on ImageNet is more effective than pre-training on medical image dataset, MIMIC-CXR, despite the latter being more relevant to the downstream task [32]. This indicates that feature learning using SSL benefits from large-scale training data (ImageNet with 1.2M images vs. MIMIC-CXR with 377K images).

### 4.4. Cross-Validation

In order to provide single-model performance for comparison against other models, the models discussed in previous sections use regular training. However, to further verify the efficacy of our full proposed TRUDLMIA framework, we conduct 5-fold cross-validation during fine-tuning on the COVIDx 8B dataset, as cross-validation allows us to estimate the performance of our general framework as opposed to the performance of single models. We test all five models on the test set to collect a range of results (see Table 10), giving us a robust estimate of the performance of TRUDLMIA on unseen data.

The results in Table 10 show stable performance across the five models produced during cross-validated training. This indicates several positive factors to the TRUDLMIA framework’s performance. First, stable performance implies strong generalization to unseen data, as performance is not dependent on a particular random split of the data. Second, the consistency of the results is indicative of robustness in terms of variability and new data; since the model is robust to changes in training data, it will likely continue to perform well even as new data comes in or if the data have some inherent variability. Finally, our stable cross-validation results show low sensitivity to partitioning, such that the model’s performance is not highly sensitive to the way the data are partitioned for training and validation, making it less likely for model performance to be skewed by the particular selection of data used for training.

### 4.5. Visual Interpretation of Results with Feature Clustering

We perform an extension of the previous ablation study with each module of the TRUDLMIA framework by computing feature embeddings for the 400 testing images in the COVIDx dataset. We then apply the *t*-SNE [44] to their embeddings, which are model outputs immediately before fully connected layers, as these outputs hold the most semantic information about the original images as well as the classes they belong to. These *t*-SNE results using a 2D mapping are plotted in Figure 2.

While all three *t*-SNE plots are able to produce adequate clusters of positive and negative samples, there are clear improvements resulting from the addition of each module, corresponding to their intended effects. The addition of the SSL module makes the clustering much more apparent than just SL, which implies an improvement in feature learning, which is the intended purpose of the SSL module. Likewise, the addition of AUC results in tighter clusters. which indicates better model confidence in terms of what class a given sample belongs to, which in turn means better trustworthiness.

### 4.6. Visual Interpretation of Results with Grad-CAM Heat Maps

We produce Grad-CAM [45] visualizations of model decision-making to further extend the previous ablation with each module of the TRUDLMIA framework. The regions in an input image that a CNN focuses on during classification are highlighted by computing the gradients of the class score with respect to the feature maps of a target layer, weighting and summing the feature maps, applying a ReLU activation, and overlaying the resulting heatmap on the original image. The last convolutional layer of the last block in our model is chosen in order to preserve spatial information and capture high-level semantic features that are strongly related to the final classification decision.

We chose two positive data samples from the test dataset. The first patient’s X-ray (Figure 3a) shows increased interstitial markings at the base of the right lung, denoting a possible early infectious process. The second patient’s X-ray (Figure 3e) shows increased interstitial markings throughout both lungs with some airspace opacities present in the left lower lobe. It should be noted that these are not radiologist interpretations or a full radiological report of the X-rays, but only an interpretation of the lung space area performed by a clinician (one of the co-authors) performed during a clinical care episode.

All three ablations are able to produce Grad-CAM visualizations that are able to approximately identify lung areas. We note that SL models are prone to making more mistakes in this regard, highlighting areas outside of the lung (shoulders, arms), which can be fixed by lung segmentation [46]. The addition of the SSL module results in better recognition of the lung space, which corresponds to the intended purpose of the SSL to facilitate better feature learning, and adding the AUC maximization module causes highlighted regions to be even more restricted in the lung area. However, while there is improvement in recognition of the general lung space through the ablations, the quality of heatmaps is not consistent in terms of pointing to an area containing a defining feature for COVID-19 pneumonia. In some cases, the SL or SL+SSL models make better predictions in terms of identifying pneumonia features while also erroneously highlighting areas that are clearly outside of the lungs.

## 5. Discussion

The presented framework does have certain limitations tied to the intrinsic characteristics of the selected methods. For instance, the high computational demand associated with self-supervised pre-training poses challenges when applied to domain-specific images with restricted computational resources. AUC maximization-based loss functions are also sensitive to model changes [10], which can lead to less stability during training.

However, modules of the TRUDLMIA framework are designed to supplement each other, minimizing the effect of these limitations. The reason for incorporating a supervised learning component was to decrease the necessity for extensive self-supervised learning, reducing the computational load required. The SL + SSL pre-training modules also reduce the amount of AUC maximization-based fine-tuning, minimizing the training instability.

Another potential weak point lies in the innovative nature of deep AUC maximization as a surrogate loss function. While deep AUC maximization is appropriate for the problem at hand due to its effectiveness in dealing with class imbalance, its novelty means that its properties have not yet been extensively studied. During training, we observed validation loss curves that were relatively unpredictable compared to the loss curves of traditional. Although this does not inherently undermine the method, the unpredictability of these loss curves reduces interpretability and might complicate the identification of issues arising during the training/validation phases.

Despite the limitations discussed, the empirical results underscore the substantial practical benefits and broad applicability of our framework. Even when compared with networks that have been expressly designed for the same tasks, our TRUDLMIA framework demonstrates superior performance. Moreover, even marginal improvements bear significance in the realm of medicine, potentially leading to direct enhancements in patient care and quality of life.

## 6. Conclusions

In computer-aided diagnosis, deep learning has demonstrated great potential when integrated with medical imaging; however, the adoption thereof is challenged by limited training data, imbalanced data sets, and low trust in models. To combat these issues, we propose TRUDLMIA, a modular deep learning framework that can produce trustworthy and high-performing models for medical image analysis.

The framework comprises three modules: large-scale supervised learning (SL) and self-supervised learning (SSL) for pre-training, as well as transfer learning-based fine-tuning with AUC margin loss. Large-scaled supervised pre-training reduces the data-hungriness and computational load of later modules. Self-supervised learning continues to address data-hungriness through efficient feature learning and reducing the need for labeled data. Finally, transfer learning with AUC margin loss prioritizes the minority positive class, dealing with the class imbalance issue. The modules also work to supplement each other’s weaknesses. Supervised pre-training reduces the high computational load of self-supervised learning, and the SL + SSL pre-training modules provide a strong backbone for fine-tuning with AUC margin loss, improving stability.

Through highly successful assessments on a benchmark dataset, we show the effectiveness of the TRUDLMIA framework for medical image analysis. Using the COVIDx dataset, models trained through the framework surpass traditional supervised models, including the state-of-the-art COVID-Net CXR-3. Using the RSNA Pneumonia Challenge dataset, the trained model achieved top performance when compared to other notable models. On the SIIM-ISIC Melanoma Classification Challenge, TRUDLMIA achieves competitive performance even when competing with models that have advantages through external data, data augmentations, and ensembling. Our top-performing high-trust models for these tasks will be made publicly available.

Moreover, ablation studies are conducted and verified to show the improvement of our framework with the addition of each module, showing significant improvements in precision, sensitivity, and trust score. We also demonstrate the stability of the framework by adding cross-validation to the fine-tuning phase, which results in a performance range that can be used to evaluate the framework methodology in general instead of single models, which may be high-performing due to lucky randomization during training. Our cross-validation experiment shows stable high-performance results, demonstrating the robustness of the framework in terms of data variability and partitioning [47]. Finally, visual interpretations of our results are shown as an extension of the aforementioned ablation study; *t*-SNE visualizations [48] show the improvement of feature learning with the addition of each module, while Grad-CAM [45] visualizations give insight to model decision making. The validity of our visual interpretations is confirmed by a clinician.

The proposed framework is by no means perfect, but we hope the TRUDLMIA can contribute to the fight against pandemics, including post-acute COVID syndrome (long COVID) [49,50], and establish a viable path for future ones. In our future work, we are interested in investigating the use of Vision Transformers [51] to replace the CNNs used in the TRUDLMIA framework. Furthermore, this work focuses on methods to deal with limited data by maximizing feature learning effectiveness with pre-training and transfer learning; it would also be worthwhile to explore other methods such as data augmentation, model ensembling, and few-shot learning [52]. Furthermore, the use of generative AI [53] can be explored for increasing dataset size, as well as its impact on model trust.

## Figures and Tables

**Figure 1 sensors-23-08122-f001:**
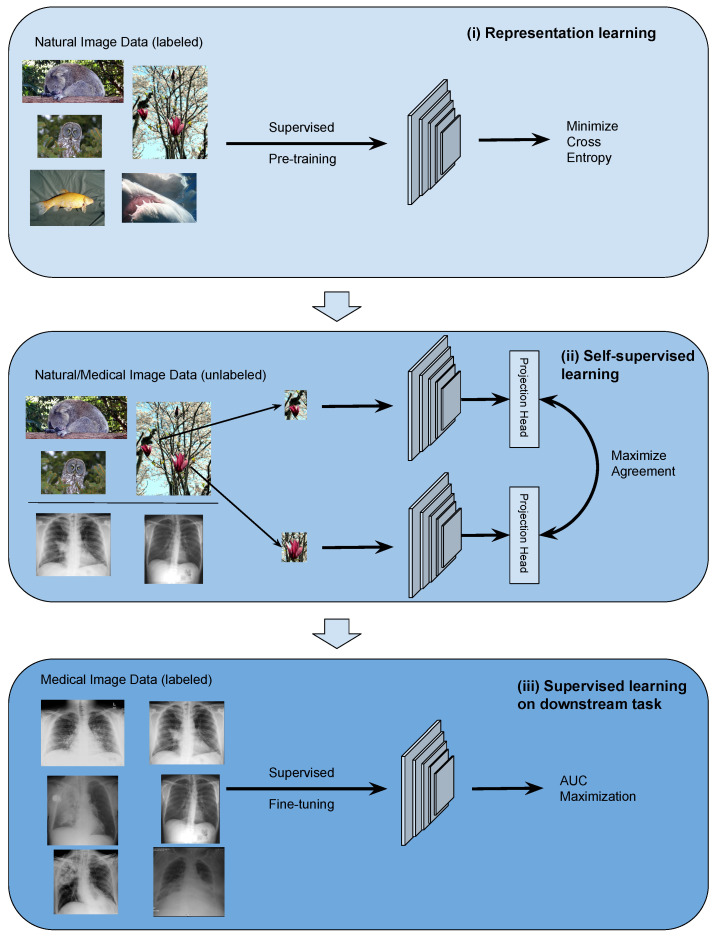
Overview of the proposed deep learning framework: (**i**) conduct general representation learning on large-scale natural image data sets using supervised learning, (**ii**) conduct refined representation learning using natural/medical image data sets with self-supervised learning, and (**iii**) conduct fine-tuning on downstream data set with supervised learning through AUC maximization.

**Figure 2 sensors-23-08122-f002:**
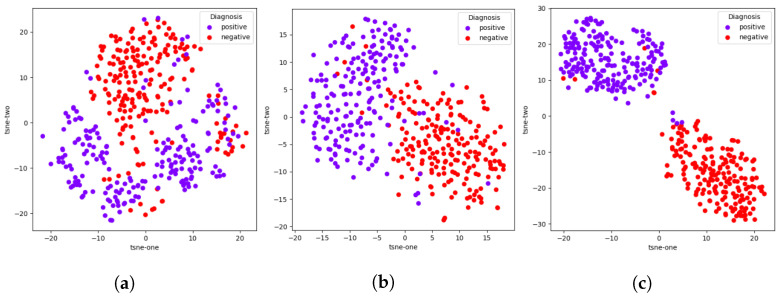
Model ablation with *t*-SNE plots of embeddings of model outputs on COVIDx dataset. (**a**) SL, (**b**) SL + SSL, (**c**) SL + SSL + AUC.

**Figure 3 sensors-23-08122-f003:**
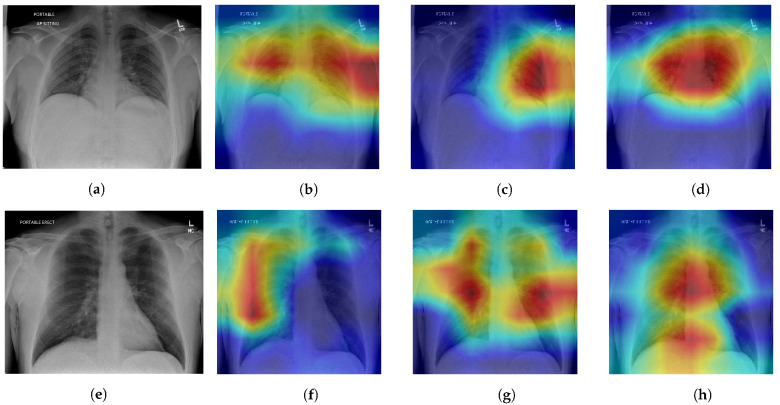
Model ablation with Grad-CAM visualizations on positive patient samples. Cooler colors (blue, green) indicate lower importance and warmer colors (red, yellow) indicate higher importance. (**a**) Original, (**b**) SL, (**c**) SL + SSL, (**d**) SL + SSL + AUC, (**e**) Original, (**f**) SL, (**g**) SL + SSL, (**h**) SL + SSL + AUC.

**Table 1 sensors-23-08122-t001:** Data split for COVIDx 8B images.

Split	Negative	Positive	Total
Train	13,793	2158	15,951
Test	200	200	400

**Table 2 sensors-23-08122-t002:** Data split for RSNA Pneumonia Challenge images.

Split	Negative	Positive	Total
Train	20,130	10,097	30,227
Test	Unknown	Unknown	4527

**Table 3 sensors-23-08122-t003:** Data split for SIIM-ISIC Melanoma Classification images.

Split	Negative	Positive	Total
Train	32,542	584	33,126
Test	Unknown	Unknown	10,982

**Table 4 sensors-23-08122-t004:** Model performance and trust scores for COVID-Net CXR-2, COVID-Net CXR-3, and COVID-CXR-SSL (best results highlighted in bold).

Architecture	Precision		Sensitivity		Trust
	**Pos.**	**Neg.**	**Pos.**	**Neg.**	
COVID-Net CXR-2	0.970	0.955	0.956	0.970	-
COVID-Net CXR-3	0.990	0.975	0.975	0.990	-
COVID-CXR-SSL (ours)	**1.000**	**0.980**	**0.980**	**1.000**	**0.964**

**Table 5 sensors-23-08122-t005:** Sensitivity, positive predictive value (PPV), and accuracy of RSNA-Pneumonia-SSL on the test data from the RSNA Pneumonia Challenge dataset in comparison to other networks (best results highlighted in bold).

Model	Sensitivity	PPV	Accuracy
SE-ResNet-50 [38]	0.440	0.909	0.680
CheXNet [39]	0.500	**0.926**	0.730
CBAM [40]	0.680	0.773	0.740
MEDUSA [37]	0.820	0.837	0.830
RSNA-Pneumonia-SSL (ours)	**0.823**	0.913	**0.873**

**Table 6 sensors-23-08122-t006:** AUC score of TRUDLMIA-Melanoma on the test data from the SIIM-ISIC Melanoma Classification dataset in comparison to other networks (best results highlighted in bold). Ext. Data = External Data and Data Aug. = Data Augmentations. The ✓and ✗ symbols indicate whether or not a method was used respectively.

Model	Ext. Data	Data Aug.	Ensemble	AUC
EfficientNet-Ensemble [43]	✓	✓	✓	0.9411
DenseNet-201 [42]	✗	✓	✗	0.9250
EfficientNet-DAM [10]	✓	✓	✓	0.9423
All Data Are Ext (winner)	✓	✓	✓	**0.9490**
TRUDLMIA-Melanoma (ours)	✗	✗	✗	0.9426

**Table 7 sensors-23-08122-t007:** Ablation study on model performance and trust scores for different model architectures with COVIDx 8B dataset (best results are bolded).

Architecture	Precision		Sensitivity		Trust
	**Pos.**	**Neg.**	**Pos.**	**Neg.**	
SL	1.000	0.885	0.870	1.000	0.918
SL + SSL	1.000	0.939	0.935	1.000	0.952
SL + SSL + AUC	**1.000**	**0.952**	**0.950**	**1.000**	**0.954**

**Table 8 sensors-23-08122-t008:** Ablation study on model performance and trust scores for different model architectures with RSNA Pneumonia Challenge dataset (best results highlighted in bold).

Architecture	Precision		Sensitivity		Trust
	**Pos.**	**Neg.**	**Pos.**	**Neg.**	
SL	0.870	0.779	0.413	**0.971**	0.897
SL + SSL	0.873	0.818	0.548	0.961	0.921
SL + SSL + AUC	**0.891**	**0.829**	**0.621**	**0.971**	**0.949**

**Table 9 sensors-23-08122-t009:** Model performance and trust scores for different SSL plug-ins on the COVIDx 8B dataset (best results are bolded).

SSL Plugin	Precision		Sensitivity		Trust
	**Pos.**	**Neg.**	**Pos.**	**Neg.**	
MoCo (*MIMIC-CXR*)	0.995	0.896	0.884	0.995	0.909
MoCo (*ImageNet*)	0.998	0.934	0.930	0.998	0.937
SimCLR (*ImageNet*)	**1.000**	**0.952**	**0.950**	**1.000**	**0.954**

**Table 10 sensors-23-08122-t010:** Five-fold cross-validation model performance and trust scores ranges for TRUDLMIA framework on COVIDx 8B test set.

Architecture	Precision		Sensitivity		Trust
	**Pos.**	**Neg.**	**Pos.**	**Neg.**	
TRUDLMIA	1.000±0%	0.949±1.4%	0.943±0.9%	1.000±0%	0.951±1.8%

## Data Availability

In this research, the authors used COVIDx dataset and RSNA Pneumonia Challenge dataset. As a part of COVID-Net Open Source Initiative, the COVIDx dataset is publicly available at https://github.com/lindawangg/COVID-Net (accessed on 8 May 2023). The RNSA dataset is part of Kaggle challenges and available at https://www.kaggle.com/c/rsna-pneumonia-detection-challenge (accessed on 8 May 2023).

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
