# Peer review of "Towards Building a Trustworthy Deep Learning Framework for Medical Image Analysis"

_sensors, 2023, doi:10.3390/s23198122_

Round 1
Reviewer 1 Report
he authors propose TRUDLMIA, a trustworthy deep learning framework for medical image analysis, which leverages image features learnt through self-supervised learning and utilizes a novel surrogate loss function to build trustworthy models with optimal performance. On the whole, this manuscript showed better results, but many limitations need to be addressed. Here are my comments.
1. Since the selected methods in comparison study seem not the newest or the most excellent ones, it does not prove that what you propose is competitive enough. So please supplement experiments and compare your method.
2. Your thoughts on the limitations of the proposed algorithm and how it can be improved should be reflected in the manuscript.
3. Relevant research background needs to be supplemented inintroduction. And you should cite all papers you use properly.
4. Conclusions needs more in it, as it's more of an afterthought. The authors are suggested to highlight important findings and include afterthought of this work.
5. The significance of this paper is not expound sufficiently. The author need to highlight this paper's innovative contributions.
6. The number of datasets used for pre-training and fine-tuning TRUDLMIA modules is not enough, you need to supplement at least 1-2 datasets for validation, so that the Manuscript is more convincing. What’s more, generalization experiment is suggested to be add to verify the generalization ability of the proposed method.
Minor editing of English language required
Reviewer 2 Report
The paper addressed an important and practical issue. It was well written. Please find my comments below.
“Class imbalance.” Please include other techniques to deal with imbalanced data, such as oversampling etc.
“Related work”: consider rewriting the last sentence in the first paragraph. It was not very clear
Tables 1 and 2. Please justify why data was split into training and testing in the presented proportions. The test sets seem quite small compared to train sets. In addition, positive instances were far less in the train sets while the test sets had balanced positive and negative instances. Please explain the logic behind such data split.
Line 181: Please describe how the random samples were obtained. Did you use stratified sampling to sample positive and negative instances separately? Please justify why cross validation was not used.
Line 191: Table 3 did you report trust scores.
Line 202: Explain why images of different resolution were used and how it might have caused the difference in model performance.
Section 4.1 Consider rewriting the first two paragraphs. There was repeated information and the flow can be improved.
Table 4: It’s worth discussing why your model produced both high sensitivity and PPV while CheXNet had higher PPV and low sensitivity.
Line 227: Consider including a table summarizing how many epochs the models were trained and adding justifications on why those stopping criteria were chosen.
“Results” Please provide computational time of the models during training. Did the new model take longer time to train? Please also add implementation detail such as the computer used and DL platform.
“Discussion” Authors should consider describing the practical benefit of the proposed study. In some results, for instance those in Table 3, the improvement seems marginal. Please also echo the three main issues outlined in introduction about AI in medical images and comment on how the proposed study addressed them effectively.
Data Availability Statement: Please consider publishing the models on github for others to use.
